# Adversarial and variational autoencoders improve metagenomic binning

Pau Piera Líndez [1], Joachim Johansen [1], Svetlana Kutuzova[1,2], Arnor Ingi Sigurdsson[1], Jakob Nybo Nissen [1✉] & Simon Rasmussen [1,3✉]

Assembly of reads from metagenomic samples is a hard problem, often resulting in highly fragmented genome assemblies. Metagenomic binning allows us to reconstruct genomes by re-grouping the sequences by their organism of origin, thus representing a crucial processing step when exploring the biological diversity of metagenomic samples. Here we present Adversarial Autoencoders for Metagenomics Binning (AAMB), an ensemble deep learning approach that integrates sequence co-abundances and tetranucleotide frequencies into a common denoised space that enables precise clustering of sequences into microbial genomes. When benchmarked, AAMB presented similar or better results compared with the state-of-the-art reference-free binner VAMB, reconstructing ~7% more near-complete (NC) genomes across simulated and real data. In addition, genomes reconstructed using AAMB had higher completeness and greater taxonomic diversity compared with VAMB. Finally, we implemented a pipeline Integrating VAMB and AAMB that enabled improved binning, recovering 20% and 29% more simulated and real NC genomes, respectively, compared to VAMB, with moderate additional runtime.

[1] Novo Nordisk Foundation Center for Protein Research, Faculty of Health and Medical Sciences, University of Copenhagen, Copenhagen N 2200, Denmark.
[2] Department of Computer Science, University of Copenhagen, DK-2100 Copenhagen Ø, Denmark. [3] The Novo Nordisk Foundation Center for Genomic Mechanisms of Disease, Broad Institute of MIT and Harvard, Cambridge 02142, USA. ✉email: jakob.nissen@cpr.ku.dk; simon.rasmussen@cpr.ku.dk

It has been estimated that there are about one trillion ($10^{12}$) species of microbes on Earth and that the large majority of these have yet to be discovered[1]. Decades ago, the standard approach to studying novel microbes was to isolate and cultivate them in the lab[2]. However, microorganisms establish and rely upon complex ecosystems that are not feasible to replicate in ex-natural environment conditions. This so-called "cultivation bottleneck" limits culturing approaches to studying microbes[1]. In contrast, metagenomics enables culture-free microbial diversity characterization by analysing the entire set of genomes present in a given environmental sample[3]. Unfortunately, despite advances in sequencing throughput and bioinformatic tooling, reconstructing high-quality genomes from short-read sequenced metagenomics samples remains challenging[4].

In particular, it is still not feasible to assemble reads from shotgun sequences to contigs that each cover whole original source genomes; instead, recovered genomes are often fragmented into many smaller contigs[5]. To mitigate this limitation of assemblers, contigs can be clustered into bins that represent their source genomes in a process called binning.

Hundreds of thousands of Metagenome Assembled Genomes (MAGs) which have previously been binned are publicly available, allowing detailed investigations of diverse microbial communities[6–8]. Despite the fact that several binners such as MetaBAT2, MaxBin 2.0, CONCOCT, and Canopy have been developed, binning performances are still far from optimal[9–13]. Recently we have developed a deep learning-based method, called VAMB, that leverages variational autoencoders (VAEs) to obtain superior performance compared to previous reference-free binners[14]. VAEs are composed of an encoder that transforms the input features into a latent distribution, and a decoder that samples from the distribution and attempts to reconstruct the input from the sample. VAEs are widely used due to their ability to represent a complex feature space into a continuous latent distribution, and their inherent sampling process makes them suitable as generative models[15].

VAMB uses a VAE to integrate input contig abundances and tetranucleotide frequencies (TNF) to a common latent representation that can be clustered to yield bins. The regularisation of the latent space is done using Kullback-Leibler divergence with respect to a prior distribution, in VAMB's case the Gaussian unit distribution[14]. However, another autoencoder framework is the adversarial autoencoder (AAE) where the regularisation of the latent space is achieved using another neural network, hence the name "adversarial"[16]. Previous work applying AAEs to images showed that AAE models could generate latent representations with sharper and better-confined clusters compared to VAEs[16]. We, therefore, hypothesised that the application of an AAE for metagenomics binning could improve on clustering of near-complete genomes from the latent space. Furthermore, the original AAE implementation used an additional categorical latent space alongside the continuous one and showed that the model learned to cluster the input by assigning each cluster to a categorical class[16,17]. We hoped that when applying AAEs to metagenomic sequences, the AAE would likewise learn to assign each genome into a single categorical class in the categorical latent space.

Here, we present Adversarial Autoencoders for Metagenomic Binning (AAMB), an extension of our original VAMB program. AAMB leverages AAEs to yield more accurate bins than VAMB's VAE-based approach. We apply AAMB to both synthetic and real metagenomic benchmark datasets and show that more high-quality genomes are recovered using AAMB compared to using VAMB or other binners and that the extra genomes expand the taxonomic diversity of recovered genomes. We also present a method for automatically merging VAMB and AAMB, and show

that the resulting ensemble method AVAMB is superior to both VAMB and AAMB while requiring nothing extra from the user other than a moderate increase in compute power.

## Results

**An adversarial autoencoder for metagenomics binning.** Inspired by the original AAE implementation, the AAE in AAMB uses both a continuous (termed z) and a categorical (termed y) latent space (Fig. 1a). Therefore, AAMB is able to extract bins both by clustering z like VAMB does[14], and by extracting the bin label directly from y. This resulted in two sets of bins, AAMB(z) and AAMB(y), respectively. When we investigated the structure of the AAMB z space when applied to the CAMI2 short-read human "toy" datasets (see Methods), we found that distances between contigs from the same genome tended to be smaller than distances between contigs from different genomes for all benchmark datasets (Supplementary Fig. 1), a prerequisite for clustering into genomes. AAMB's z space was more compact than the VAMB latent space, which we believe was due partly to AAMB's ability to encode information in y. When clustering z, it yielded somewhat worse bins than VAMB, giving a total of 7% fewer NC genomes compared to VAMB across the CAMI2 and MetaHIT datasets (Fig. 1c). The clusters of AAMB(y) were likewise inferior to VAMB, giving on average 39% fewer NC genomes. The relative performance of AAMB(z) vs AAMB(y) was dataset dependent; AAMB(z) outperformed AAMB(y) on the CAMI2 Airways, Gastrointestinal, Oral, Skin, and Urogenital datasets, reconstructing between 47–102% more NC genomes. On the contrary, AAMB(y), outperformed AAMB(z), on the MetaHIT dataset, by reconstructing 164% more NC genomes (Supplementary Table 1). Interestingly, the MetaHIT dataset has been difficult for all the binners we have tested, including MetaBAT2, MaxBin2.0 and Canopy. This suggested to us that the z and y spaces contained encodings of different subsets of the total information in the input data.

**The continuous and categorical latent space encodes different information.** The original AAE paper showed how the y and z space primarily encoded high-level variance ("class") and low-level variance ("style"), respectively. We hypothesised that AAMB behaved similarly, i.e. AAMB(y) would cluster contigs at a higher taxonomic rank than AAMB(z), which would imply that AAMB(y) might conflate lower taxonomic ranks. To test this, we measured the taxonomic distance between randomly selected contig pairs from the same cluster in AAMB(y) and AAMB(z) (Supplementary Fig. 2). We found that AAMB(y) conflated higher ranks more often than AAMB(z), and AAMB(z) conflated strains more often than AAMB(y), which did not support our hypothesis. The hypothesis further implied that while most AAMB(z) clusters would have high strain-level purity, multiple AAMB(y) clusters representing different higher taxonomic ranks could sometimes be mapped to the same AAMB(z) space, causing some AAMB(z) clusters to be the union of otherwise pure strains from disparate high taxonomic ranks. If that was the case, it would imply that some contig pairs from the same AAMB(z) cluster would be from wildly different clades, but we did not observe this (Supplementary Fig. 3). Further when splitting AAMB(z) clusters by their y label to decontaminate this potential contamination, we found the resulting bins were no better (Supplementary Table 2), contrary to our hypothesis. We thus concluded that neither AAMB(z) nor AAMB(y) were redundant with respect to each other, nor did z only capture intra-y variance, but instead the two latent spaces learned complementary information. Therefore, we decided to explore the intersection and differences in genome reconstruction between VAMB, AAMB(z),

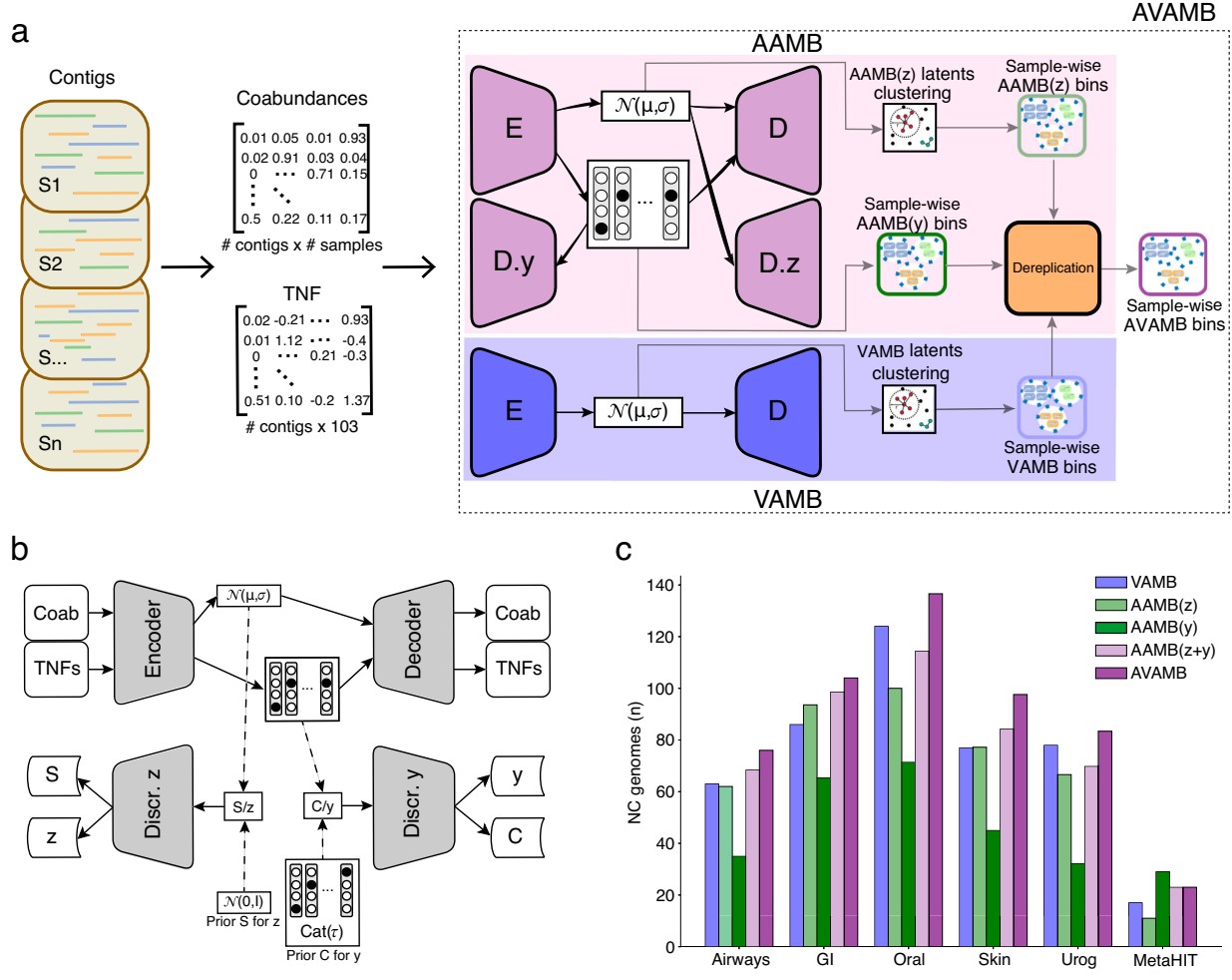

**Fig. 1 AVAMB and AAMB workflow overview, adversarial autoencoder model schematic representation, and AVAMB and AAMB performance across the benchmark datasets. a** AAMB workflow overview. Tetranucleotide frequencies and abundances across samples are extracted per contig and input to the AAMB encoder. After training, latent representations z and y are retrieved. Then, the VAMB clustering algorithm was applied to generate clusters from the z latent representation, and cluster labels were taken directly from y. Finally, bins from z and y are deduplicated to the final AAMB clusters. These can then potentially be integrated with VAMB generated clusters, in that case named AVAMB. Dark arrows represent forward propagations, grey arrows represent clustering and de-replication steps performed after training AAMB and VAMB. **b** Adversarial autoencoder model overview. The encoder-decoder was optimised to reconstruct the input contig features from the regularized latent representations z and y. Regularisation is achieved by adversarial competition between the discriminators and the encoder, enforcing the latent encodings to stay close to their prior distributions. Dark arrows represent forward propagations. Dashed arrows represent sampling processes from the latent and priors. **c** Number of distinct NC genomes reconstructed from the six benchmark datasets for VAMB (blue), AAMB(z) (light green), AAMB(y) (dark green), AAMB(z + y) (light purple), AVAMB (dark purple). GI Gastrointestinal, Urog Urogenital.

and AAMB(y) (Fig. 2a). We found that NC genomes reconstructed by AAMB(y) had a Jaccard index of 0.40 and 0.47 to the NC genomes reconstructed by AAMB(z), and VAMB, respectively (see Methods), while the Jaccard index of NC bins from AAMB(z) versus VAMB was higher at 0.64. Hence AAMB(z)'s genomes were more similar to VAMB's than to AAMB(y)'s. This was not surprising, as the continuous Gaussian latent space of VAMB is more akin to the similarly structured z.

**Combining the latent spaces AAMB(y) and AAMB(z).** Because AAMB(z) and AAMB(y) reconstructed different sets of genomes, we developed a technique to de-replicate the genome sets reconstructed from the two latent spaces. Briefly, we assessed bin quality with CheckM2[18] to remove low-quality bins, then for each

bin pair that we deemed to be nearly identical, we removed the lowest-scoring bin. Finally, any contigs contained in two or more bins were assigned to the bin whose CheckM2 score would be improved the most by the addition of these contigs (see Methods). When comparing this de-replication method to a popular alternative MAG de-replication tool dRep[19], our method preserved more genomes, only losing 28 NC bins in the de-replication process across all the CAMI2 datasets, compared to 40 when using dRep (Supplementary Table 3). Simultaneously, our approach produced no duplicated contigs, whereas 6 contigs remain duplicated when using dRep (Supplementary Table 4). Applying this de-replication process to the union of AAMB(z) and AAMB(y), we obtained what we termed AAMB(z + y), and found that it outperformed VAMB, reconstructing 7% more NC genomes across all CAMI2 datasets (Fig. 1c). This workflow came

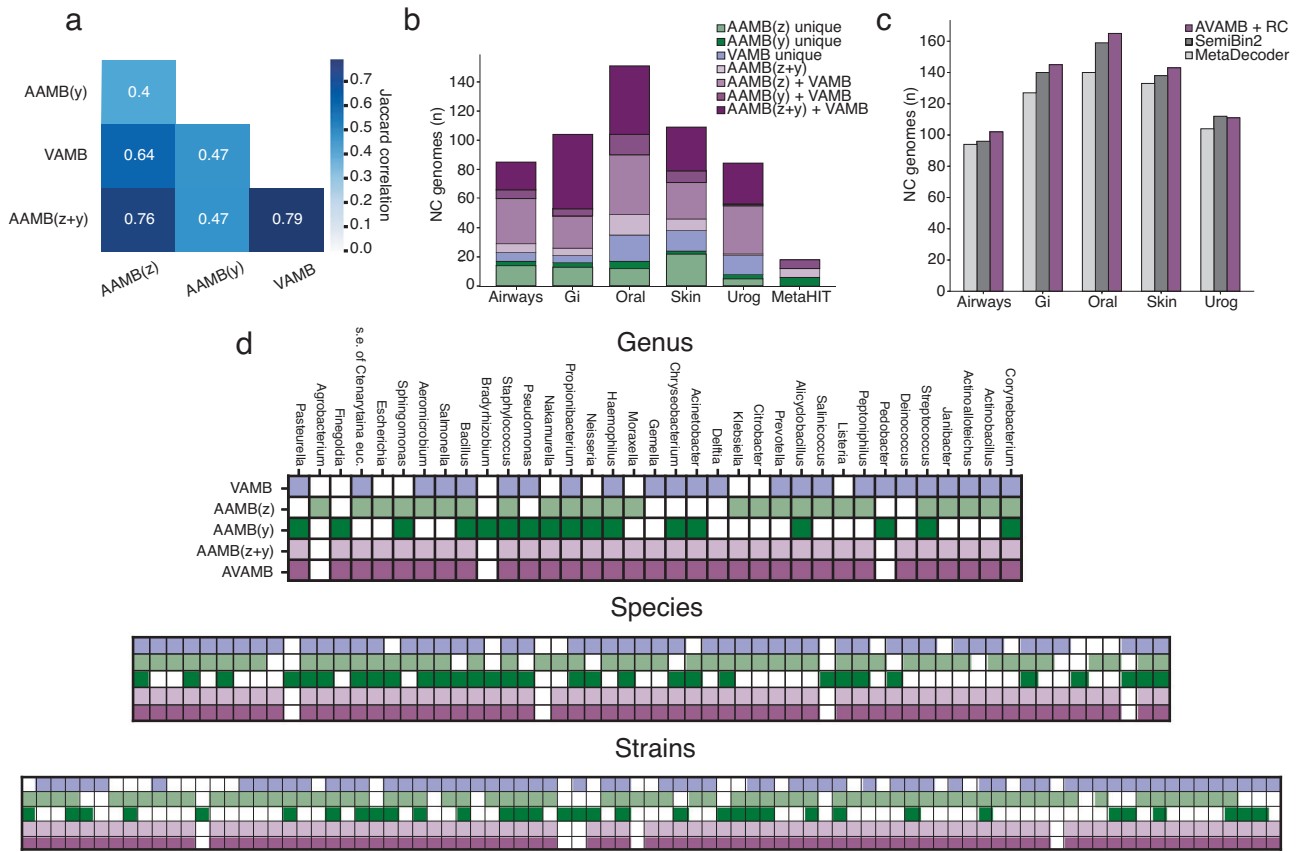

**Fig. 2 AAMB and VAMB reconstructed genomes analysis and integration with de-replication workflow. a** Jaccard correlation between the NC genomes from all benchmark datasets produced by AAMB(z), AAMB(y), VAMB, and AAMB(z + y) de-replicated bins. **b** Contribution and the intersection of AAMB and VAMB for NC genomes on CAMI2 and MetaHIT datasets. The aggregated height of each bar expresses the total number of NC genomes reconstructed by VAMB and AAMB when de-replicating. Unique: NC genomes only reconstructed by the given binner; and: NC genomes reconstructed by all the binner connected with the and operator and not by any other binner; NC: Near complete. **c** Number of distinct NC genomes reconstructed from the five CAMI2 benchmark datasets for AVAMB + RC (purple), SemiBin2 (dark grey), and MetaDecoder (light grey). GI, Gastrointestinal; Urog, Urogenital; AVAMB + RC, AVAMB NC genomes when the single-copy genes-based re-clustering from SemiBin2 was applied to AVAMB's workflow. **d** Comparison of strains, species, and genera recovered at NC level by VAMB (blue), AAMB(z) (light green), AAMB(y) (dark green), as well as AAMB(z + y) (light purple), and AVAMB (dark purple). *S.e. of Ctenarytaina euc., secondary endosymbiont of Ctenarytaina eucalypti.*

at some cost of computational power, as training and clustering time required were 1.9x and 3.4x higher when using GPU or CPU only, respectively, compared to VAMB (Supplementary Table 5). Because AAMB(z + y) had the best performance, we used it as the default AAMB workflow and will henceforth refer to it as simply AAMB.

**Combining the AAMB and VAMB framework.** We found that AAMB (i.e. AAMB(z + y)) reconstructed a different set of NC genomes than VAMB (Jaccard index of 0.79) (Fig. 2a, b, d, Supplementary Fig. 4–9), and so hypothesised that we could apply the AAMB de-replication approach to also merge VAMB bins to form a VAMB + AAMB workflow we called AVAMB. The integrated AVAMB model proved highly performant as it allowed the reconstruction of 6–35% additional NC genomes compared to using only VAMB across the benchmark datasets (Fig. 1c, Supplementary Table 6). Furthermore, a few (2-7) NC genomes were lost during de-replication of AAMB and VAMB bins. We then investigated the contribution from AAMB(z), AAMB(y), and VAMB, and found that 23-50% of the NC genomes were reconstructed by all three methods across the benchmark datasets. On the other hand, we found that up to 27% of the NC genomes in a dataset were only identified by one

method (Fig. 2b, Supplementary Table 7). We concluded that VAMB, AAMB(z), and AAMB(y), each reconstruct different genomes from the benchmark datasets and that the de-replicated union of all three methods yields better overall performance. To check if the gains of AVAMB over VAMB were simply due to the former being an ensemble method, we merged the bins of VAMB and MetaBAT2 and compared this VAMB+MetaBAT2 with VAMB + AAMB. We found that VAMB + AAMB outperformed VAMB+MetaBAT2 on 5 of the 6 benchmark datasets, although an ensemble of all three methods did best on 5 of 6 datasets (Supplementary Table 8). This implies that AAMB has better synergy with VAMB than MetaBAT2. Since AAMB and VAMB share much of their pipelines including the computation of inputs to the encoders and are provided by the same software package, VAMB + AAMB is also notably faster and easier to use than VAMB+MetaBAT2. Finally, we compared AVAMB to binners that beyond contig TNF and abundances, also leverage single-copy genes or contig taxonomic annotations for binning. Namely, we benchmarked AVAMB against SemiBin[20], SemiBin2[21], and MetaDecoder[13], and found that they reconstructed 15-36%, 15-45% and 1%–28% more NC genomes compared to AVAMB on the CAMI data sets respectively (Supplementary Table 9). In the case of SemiBin and SemiBin2, performance increase came at a higher computational cost, taking 35–57 and 14–29 times longer

than AVAMB respectively (Supplementary Table 10), whereas MetaDecoder running time was similar than AVAMB (Supplementary Table 11). Furthermore, we found SemiBin2 quite dependent on the single-copy genes-based re-clustering, given that AVAMB produced 0-32% more NC genomes on the CAMI data sets compared to SemiBin2 when single-copy genes-based re-clustering was not applied (Supplementary Table 12). In addition, when the single-copy genes-based re-clustering from SemiBin2 was applied to the AVAMB workflow, AVAMB outperformed SemiBin2 and MetaDecoder on 4/5 and 5/5 CAMI2 datasets respectively, reconstructing 21 and 68 more NC genomes along all CAMI2 datasets, respectively (Fig. 2c, Supplementary Table 13).

**AAMB outperformed and complemented VAMB on real metagenomic datasets**. The synthetic nature of the CAMI2 datasets enabled precise benchmarking, but because synthetic metagenomic datasets are not entirely realistic as they use idealised coverage distributions and read assemblies[4], we did not know if AAMB's accuracy translated to real datasets. Therefore, we ran AAMB on two real datasets, a 1000 gut microbiome dataset from Almeida et al[22]. and the Human Microbiome Project 2 Inflammatory Bowel Disease cohort with 1306 longitudinal samples from 90 patients (HMP2 dataset)[23], and evaluated accuracy using CheckM2 (see Methods). We note that evaluating AAMB using CheckM2 was not entirely unbiased, since AAMB de-replicate similar bins using CheckM2 scores. Nonetheless, the observed results mirrored the ones seen on the synthetic CAMI datasets. AAMB(z) generated more NC bins than AAMB(y) with 1464 and 1963 more on the Almeida and HMP2 datasets, respectively. AAMB performed better than either AAMB(z) and AAMB(y) as well as better than VAMB, reconstructing 5077 and 2715 NC genomes, an increase of 9.7% and 2.3% (Fig. 3a, Supplementary Table 14) compared to VAMB. Similarly, AVAMB did even better, yielding a total of 5733 and 3569 NC bins, which corresponded to an increase of 1103 (24%) and 914 (34%) NC bins compared to VAMB. Similarly, we tried to use SemiBin and SemiBin2 on the Almeida data set, however, it did not complete within one week. From the samples that SemiBin finished (134), we estimated that it would take 52 days to complete using a GPU whereas SemiBin2 did not finish any sample within a week. In comparison, AVAMB finished in less than one day (22 h) (Supplementary Table 15). Finally, we benchmarked AVAMB against MetaDecoder on 30 randomly selected samples from the Almeida dataset. On those samples, AVAMB reconstructed 148 NC genomes, whereas MetaDecoder reconstructed 82 NC genomes according to CheckM2. Besides the higher performance, AVAMB was 2.8 faster than MetaDecoder. AVAMB NC genomes performance did not increase when the single-copy genes-based re-clustering from SemiBin2 was applied to AVAMB, producing the same number of NC genomes.

**AVAMB recovers more distinct taxa than VAMB at higher quality**. In order to investigate the nature of the additional genomes recovered by AAMB compared to VAMB, we assigned taxonomy to the NC bins recovered from the Almeida and HMP2 datasets (see Methods). We found that the additional genomes recovered by AAMB compared to VAMB increased diversity on the species level, but not higher taxonomic levels. As before, AVAMB performed better than either binner alone and increased diversity further on the species level where it recovered 13% and 28% more unique species from the Almeida and HMP2 datasets respectively, compared to VAMB. Unlike AAMB, AVAMB also increased diversity on the genus level, where it reconstructed 5.5% and 14% more genera than VAMB (Fig. 3b, c, Supplementary

Tables 16, 17). To gain a better overview of which clades were improved using AVAMB compared to VAMB, we created a phylogenetic tree of core genes from NC bins recovered by AVAMB (Fig. 4). In the figure, we marked bins that were uniquely recovered by AVAMB and not VAMB, and also marked bins that were recovered by both binners, but where the bin recovered by AVAMB was of better quality. AVAMB improved the quality of 2091 NC bins also reconstructed by VAMB (Supplementary Table 18). We found that AVAMB recovered more bins across the entire tree with no apparent biases towards any particular clades, but that improved bins tended to cluster together in smaller clades across the tree. When benchmarking AVAMB against MetaBAT2, we found that AVAMB reconstructed 27% and 12% more unique species and genus than MetaBAT2 from the Almeida dataset (Supplementary Table 19). Finally, AVAMB performance was also higher than MetaDecoder on 30 randomly selected Almeida samples, yielding 48% and 25% more unique species and genera respectively (Supplementary Table 20).

## Discussion

We present AAMB, an adversarial autoencoder for metagenomic binning. AAMB is a probabilistic deep learning model that attempts to integrate and denoise contig features into two latent spaces. AAMB gave better results than VAMB, a state-of-the-art reference-free binning method. We found that AAMB bins excellently complemented VAMB bins in number, bin quality, and taxonomic diversity. Thus, de-replicating the union of VAMB and AAMB's bins was found to maximise microbial genome recovery in both synthetic and real metagenomic datasets.

De-replicating multiple binnings of the same contig set is not straightforward, but is simpler than dRep's goal of de-replicating arbitrary genomes, because the former case involves duplication of the exact same contigs which is trivial to detect. Furthermore, pairs of imperfectly binned MAGs may have a nucleotide identity profile across the genome that is biologically unrealistic, with some large sections that are nearly identical, and other large sections with low nucleotide identity, a circumstance which dRep might not be built for. Hence, we believe the reason our de-replication method was more accurate than dRep was that our method was designed for the more narrow problem of de-replicating MAGs derived from the same set of contigs.

The increase in complexity of AAMB with respect to VAMB is not costless. We found that the training and clustering steps of the pipeline of AAMB were 1.8 to 2.7 times slower on average compared to VAMB's when running on a GPU and CPUs, respectively. Furthermore, de-replication of AAMB and VAMB bins per sample must be accomplished if optimal bin integration is to be obtained, extending the total runtime by up to 4.2 min per sample for the largest datasets evaluated.

Like VAMB, AAMB leverages a multi-sample approach[14], and so is able to use its advantages, including more co-abundance signal from the same contigs, increased density of contig clusters by observing homologous contigs in multiple samples, and increased binning speed. Even while being slower than VAMB, AVAMB ran 28 times faster on the Almeida dataset than running MetaBAT2 on each sample and reconstructed 92% more NC bins. Expectedly, AVAMB reconstructed fewer NC genomes on the CAMI datasets than the semi-supervised binner SemiBin, SemiBin2 and MetaDecoder. However, this was also expected as being semi-supervised they use database information to help bin the contigs at the cost of a substantial runtime increase. On the contrary, AAMB and VAMB are unsupervised in their approach to generating the genome bins. Furthermore, when a similar level

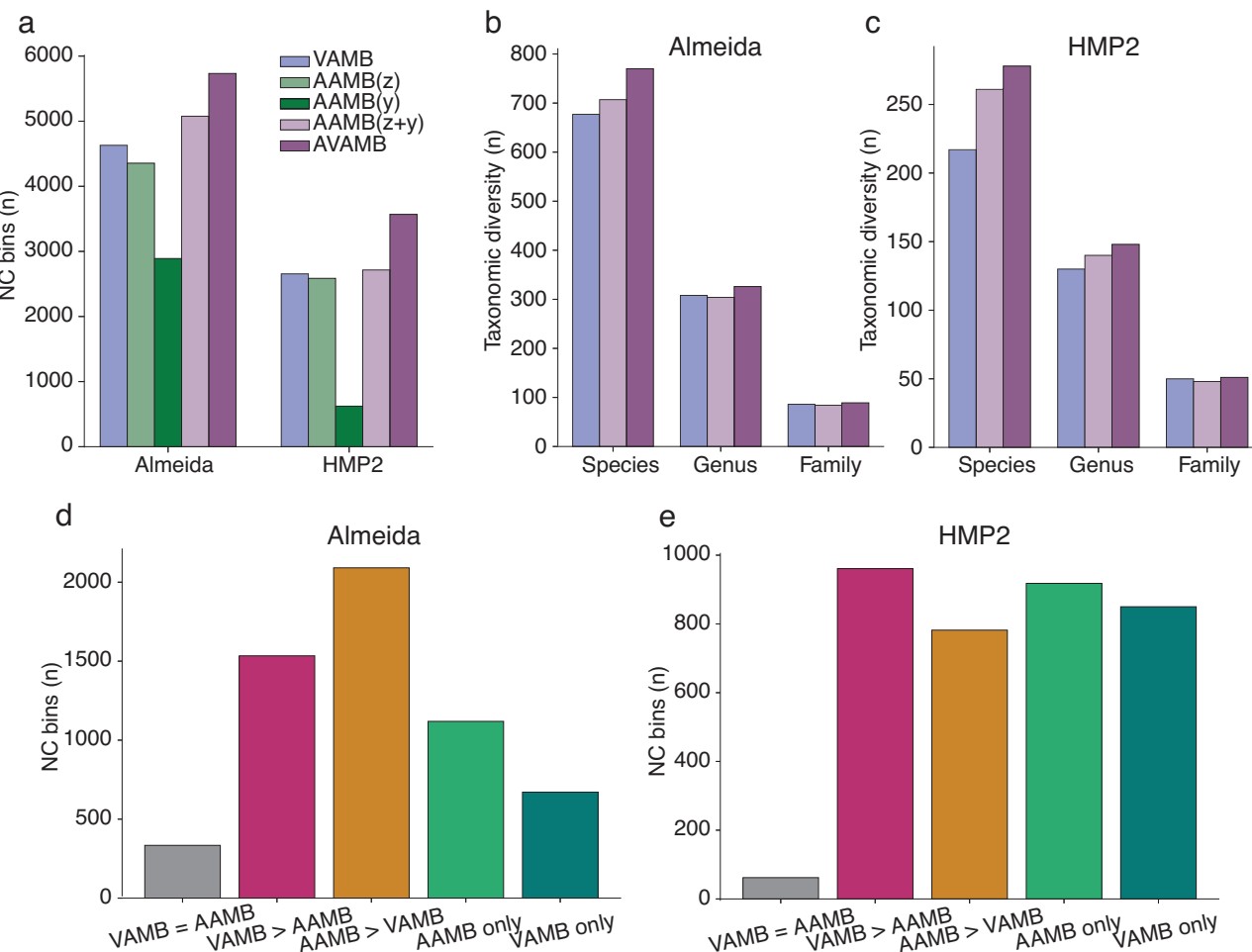

**Fig. 3 AAMB performance on real datasets. a** Number of NC bins reconstructed from the Almeida and HMP2 datasets for VAMB (blue), AAMB(z) (light green), AAMB(y) (dark green), AAMB(z + y) (light purple), AVAMB (dark purple). **b** Number of taxa with at least one NC bin genome from VAMB, the de-replicated set of bins generated by AAMB, and the de-replicated set of bins generated by AVAMB from the Almeida dataset. NC bins were classified with GTDB-Tk. Taxa reconstructed by VAMB (blue), taxa reconstructed by AAMB (light purple), and taxa reconstructed by both AAMB and VAMB (dark purple). **c** Same as B for the HMP2 dataset. **d** Almeida NC bins quality comparison between VAMB and AAMB. VAMB and AAMB bins were considered equal if belonging to the same sample with 100% identity over at least 75% of the smallest bin. NC: Near complete. V = A: VAMB and AAMB NC bins with exact same score. V > A: VAMB NC bins with a higher score than AAMB NC bins, A > V: AAMB NC bins with a higher score than VAMB NC bins, V unique: VAMB NC bins not reconstructed by any AAMB NC bin at the selected identity settings, A unique: AAMB NC bins not reconstructed by any VAMB NC bin at the selected identity settings. **e** Same to D for the HMP2 dataset.

of supervision was used in AVAMB, AVAMB showed superior performance on 4/5 CAMI datasets. Further improvement to AAMB, VAMB, and AVAMB could be done by implementing self or semi-supervised learning in the binning step itself in an efficient manner.

AAMB, the de-replication workflow, and AVAMB have been implemented using Python 3.9.16 and Snakemake[24], and are shipped with the newest release of VAMB. From a software user perspective, it thus presents minimal differences with respect to the normal VAMB workflow.

## Methods
**Overview of AAMB**. The AAMB workflow consists of the following main steps (Fig. 1a). First, Tetra Nucleotide Frequencies (TNF) and per sample co-abundances (Coab) are extracted from the contigs and BAM files of reads mapped to contigs, and input to the AAMB model as a concatenated vector. The AAMB model encodes a continuous latent space (z) and a categorical latent space (y) and reconstructs the input from these two as the output. After training, two sets of contig clusters will be generated, one set

from continuous latent space (z clusters) and another set from the categorical latent space (y clusters). The z space is clustered using the clustering algorithm of VAMB[14]. The y clusters are implicit from the one-hot categorical latent space and can simply be extracted as the categorical vector. Therefore, all contigs are present twice, both in the y and z output. The two sets were then split by sample of origin to per sample-specific bins using the principle of multi-split binning from the VAMB framework. Finally, the bins were filtered based on CheckM2[18] scores and then de-replicated as described below.

**Datasets**. We used the same synthetic benchmark datasets used in VAMB. For hyperparameter tuning, we used the short read Airways ($n = 10$), Oral ($n = 10$), and Urogenital ($n = 9$) from the Critical Assessment of Metagenome Interpretation (CAMI2) short-read 'toy' human datasets[4]. Furthermore, we used the MetaHIT 'error-free' dataset as a training set as well[25]. The remaining two CAMI2 toy human datasets, CAMI2 Gastro-intestinal ($n = 10$), and CAMI2 Skin ($n = 10$) were used for model validation. We further evaluated the methods on a

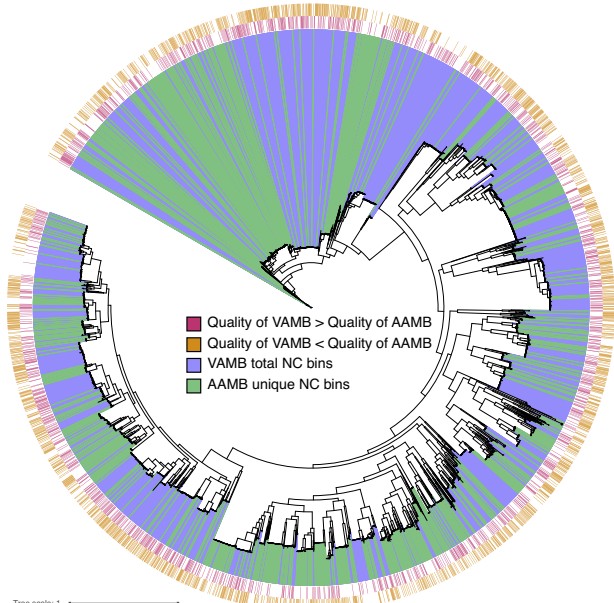

**Fig. 4 Maximum-likelihood phylogenetic tree of the NC bins reconstructed by VAMB and AAMB from the Almeida dataset.** GTDB-tk was used to generate the multiple sequence alignment, IQ-tree to produce the tree, and iTOL to visualise the tree. Red stripe: bins reconstructed by VAMB with a higher score than AAMB, golden stripe: reconstructed by AAMB with a higher score than AAMB, blue range: bins reconstructed by VAMB, green range: bins reconstructed only by AAMB and not by VAMB.

1000-sample human gut microbiome dataset collected by Almeida et al[22]. and processed by Nissen et al[14].. Additionally, we evaluated the methods on a dataset from the Human Microbiome Project 2 (HMP2) Inflammatory Bowel Disease (IBD) cohort consisting of 1338 samples from a total of 27 healthy controls, 65 Crohn's Disease, and 38 Ulcerative Colitis patients from Lloyd-Price et al[23]. and processed by Johansen et al[26].

**Contig pre-processing.** Contig sequences were transcribed into numerical vectors following the same procedure described for VAMB[14] and summarised below. TNFs were obtained by counting frequencies of all possible tetramers not containing ambiguous bases per contig. Subsequently, TNFs were projected into a 103-dimensional orthonormal space[27]. Contig co-abundances (Coab) were calculated by counting the individual reads mapping per contig. In case a read mapped to $n$ contigs, it counted $1/n$ for each contig. For the MetaHIT dataset[25], contig abundances were defined by the original authors whereas, for the CAMI2[4], Almeida[22], and HMP2[23] datasets the contig abundances were determined as done by Nissen et al[14]. In short, contigs for each sample were merged into a catalogue, and reads from each sample were aligned using minimap2 (v.2.15r905)[28] to the catalogue. For the HMP2 dataset, we used minimap2 version v.2.6. Abundances were calculated using the program *jgi_summarize_bam_contig_depths* from MetaBAT2 (v.2.10.2)[12]. When input to VAMB and AAMB, the abundances were normalised across samples for the same contig to sum to one in order to mimic a probability distribution that a random mapping read will come from each sample. TNFs were calculated using the method as described in VAMB, i.e. projected into 103 orthonormal dimensions as originally described by Kislyuk et al[27]. and normalised by z-scaling each tetranucleotide across the contigs in order to increase the relative inter-contig variance[14]. As done in VAMB, each contig TNFs and Coab vector was concatenated into a vector

of a number of samples + 103, constituting the model input vector [TNFs, Coab]$^T$.

**Adversarial autoencoder architecture.** The AAE architecture was composed of three modules: The encoder-decoder, the $z$ latent space discriminator, and the y latent space discriminator, as done in the original work from Makhzani et al[16]. (Fig. 1b). The encoder-decoder module learns the contig features and encodes them into the z and y latent spaces. The discriminators for the z and y latent spaces restrict the latent to be similar to their priors, thus performing the regularisation of the model, which imposes a structure on the latent space that makes it clusterable. The encoder is a sequence of two dense layers with 547 units of dense layers, each with LeakyReLU activation function[29] and batch normalization[30]. The encoder is connected to the dense layers μ, σ each with 283 units parameterizing $z$, and to the y layer with 700 units. Softmax[31] activation is applied on the y latent layer to mimic a probability distribution. The decoder reconstructs the contig features using a sample of the Gaussian distribution z ~ N(μ, σ), and the y vector. Its architecture is identical to the encoder. Considering that Coab was normalised across samples to sum to one, softmax was applied to the Coab output units. The two discriminators of the z and y space, $D_z$ and $D_y$, are both networks with the same architecture as the decoder model, except without batch normalisation, where the final output layer is one single node. Each discriminator is trained to discriminate between samples taken from the latent spaces (z, y) and the priors. The prior for the $D_z$ is the unit Gaussian distribution N(μ = 0, σ = I). For $D_y$, the prior is the RelaxedOneHotCategorical[32] distribution Cat(τ), with the temperature τ = 0.15. The discriminators were optimised for a binary classification task, therefore the softmax activation function was used for the output node in order to interpret the output as a probability. All three modules are optimised with Adam[32], and are implemented using PyTorch (v.1.7.1)[32], with CUDA (v.8.0.61) being used when a GPU is available. For the results in this paper, we used an NVIDIA Tesla V100 GPU, and an Intel Xeon Gold 6230 CPU.

**Loss functions.** The two discriminators were trained to minimise the difference between their binary prediction and the ground truth using binary cross entropy (BCE) on a pair of samples from the prior and the latent space, and hence use the loss:

$$LDz = 1/2BCE(D_z(z), 0) + 1/2BCE(D_z(S \sim \mathcal{N}(0, I)), 1) \quad (1)$$

$$LDy = 1/2BCE(D_y(y), 0) + 1/2BCE(D_y(C \sim C(\tau)), 1) \quad (2)$$

Where LDz and LDy is loss of the discriminators $D_z$ and $D_y$, respectively, and $S \sim \mathcal{N}(0, I)$ and $C \sim Cat(\tau)$ are samples from the priors, namely the standard normal distribution and the Gumbel-softmax distribution[33] with temperature parameter τ.

The encoder/decoder pair's loss function is the sum of two terms: Reconstruction loss $L_{rec}$ and regularisation loss $L_{reg}$. $L_{rec}$ encourages the networks to faithfully encode the input data's information and was implemented as done in VAMB[14]. In short, the cross entropy (CE) loss was used for reconstructions of the abundances of the contigs ($A_{in}$ vs $A_{out}$) and mean squared error (MSE) for the reconstruction of the TNFs ($T_{in}$ vs. $T_{out}$). These two terms are weighted with hyperparameters $w_{coab}$ and $w_{TNF}$.

$$L_{rec} = w_{coab}CE(A_{in}, A_{out}) + w_{TNF}MSE(T_{in}, T_{out}) \quad (3)$$

$L_{reg}$ encourages the latent space to be similar to their priors and is the sum of two terms, which measure the ability of the discriminator to correctly identify the samples from the latent spaces:

$$LRz = BCE(D_z(z), 1) \quad (4)$$

$$LRy = BCE(D_y(y), 1) \tag{5}$$

$L_{reg}$ is then calculated as the sum of LRz and LRy, weighted by the slr hyperparameter:

$$L_{reg} = (1 - slr)LRz + slr\,LRy \tag{6}$$

$$L_{reg} = (1 - slr) - LDz + slr * -LDy \tag{7}$$

The final model loss L is computed by weighing $L_{rec}$ and $L_{reg}$ using the hyperpameter sl:

$$L = (1 - sl)L_{rec} + sl\,L_{reg} \tag{8}$$

**Benchmarking**. For the CAMI2 dataset, strains, species, and genus were defined exactly by the original authors[4]. For the MetaHIT dataset, strain definition was defined as the genomes used to generate the dataset, while species and genus were defined using NCBI taxonomy of these strains[34]. Calculation of AAMB bin quality within each taxonomy clade was done as in VAMB:[14] At the strain level, AAMB bin precision and recall were computed using the *vamb.benchmark* tools for all genomes in the dataset. Here *precision* quantifies the purity of the genome in a bin, and was defined as follows for pair (B, G) of bin B and genome G:

$$Precision_{B,G} = \frac{bp\ of\ G\ covered\ by\ B\ contigs}{bp\ of\ any\ genome\ covered\ by\ B\ contigs} \tag{9}$$

Similarly, recall quantify genome retrieval and was defined for bin B and genome G as follows:

$$Recall_{B,G} = \frac{bp\ of\ G\ covered\ by\ B\ contigs}{Total\ bp\ in\ G} \tag{10}$$

For a higher taxonomic clade L (e.g. species or genus), $Recall_{B,L}$ and $Recall_{B,L}$ was defined by

$$Recall_{B,L} = \max_{G \in L} Recall_{B,G} \tag{11}$$

$$precision_{B,L} = \sum_{G \in L} precision_{B,G} \tag{12}$$

Binner performance is then given as the number of genomes reconstructed at some recall/precision threshold (typically 0.9/0.95) in any bin. SemiBin[20] v0.7.0 was run using the *multi_easy_bin* mode. SemiBin2[21] v1.5.1 was run using the *multi_easy_bin* with the *--self-supervised* flag. In the run time comparison, we used 20 CPU, 1 GPU, 20 GB of RAM, for VAMB, AAMB, and AVAMB, and 1 GPU, 20 CPU, 150 GB of RAM for SemiBin and SemiBin2. SemiBin2 without re-clustering was run using the *multi_easy_bin* with the *--self-supervised* and *--write-pre-reclustering-bins* flags. MetaDecoder[13] v1.0.17 was run using the *coverage, seed* and *cluster* commands, and we used 20 CPU, and 60 GB of RAM.

**Hyperparameter searches**. We did two random searches to select the hyperparameters of the AAMB model. During the first random search (RS1), the evaluated hyperparameters were selected to optimise the binning performance of the normally distributed latent space z (Supplementary Fig. 10). Because adversarial models can be more unstable during training compared to variational models, the categorical latent y was not included. We first evaluated the reconstruction/regularisation scaling factor sl, and number/shape of the encoder/decoder hidden units, as these are important for stable competition between the encoder/decoder and the discriminators. Then, we added the categorial y latent, and optimised the related parameters slr and τ in a second random search (RS2), as these determine the model complexity for learning and encoding, respectively. (Supplementary Fig. 11). For each iteration, the hyperparameters were randomly sampled

within a given range, training the model independently on each of the training datasets: CAMI2 Airways, CAMI2 Urog, CAMI2 Oral and MetaHIT. Performance was evaluated based on the number of reconstructed genomes with precision above 0.9 and recall above 0.9. The final hyperparameters were sl = 0.0964, slr = 0.5, τ = 0.1596, with encoder and decoders of two hidden layers with 547 hidden nodes per layer, 283 nodes in the latent z layer, and 700 nodes for the categorical y layer. However, the categorical y layer dimension could be always adjusted depending upon the estimated taxonomic diversity. Thus, we decided to increment the y layer size for the Almeida and HMP2 datasets since Nissen et al[14]. reported a higher diversity with respect to the CAMI2 and MetaHIT datasets.

**De-replicating genomes between latent spaces**. Because the latent spaces each encode every contig, the contigs are binned multiple times, and the same bin may be output multiple times in AAMB. Therefore, we devised a technique to de-replicate the sets of genomes. First, we ran CheckM2[18] v0.1.3 to assign completeness and contamination to all genomes and removed all genomes that were not NC (i.e. completeness >0.9, contamination <0.05). Each genome was assigned a score computed as

$$score = completeness - 5 \cdot contamination \tag{13}$$

We then identified all "near-identical" pairs of bins, where at least 75% of the smaller bin's nucleotide content was present in the larger bin. For each of these pairs, we removed the bin with the lowest score. For each contig still shared by multiple bins, we created a bin without that contig and scored it with CheckM2. The contig was then assigned the bin to which its removal would result in the largest score drop. (Supplementary Fig. 12). To compare this technique against dRep[19], we ran dRep v3.0.0 on each sample independently using default parameters. We decided to score the bins with CheckM2 instead of CheckM[35] v1.2.2 since we found that overall, CheckM2 better estimated the CAMI datasets contamination and completeness with respect the real metrics (Supplementary Fig. 13). We acknowledge that using CheckM2 both during the de-replication process and the binning evaluation might introduce some biases. However, such biases mainly apply to the bins' selection process from AAMB(z), AAMB(y), and VAMB, rather than to the bin generation process. In addition, according to the lower performance of CheckM compared to CheckM2 on bins generated from CAMI2 (Supplementary Fig. 13), we consider using CheckM for binning evaluation not optimal (Supplementary Table 21).

**AAMB and VAMB complementarity tree**. NC bins from AVAMB were de-replicated and scored (see De-replicating genomes between latent spaces sub-section). If a pair of bins, one from VAMB and the other from AAMB was identified as near-identical, then we consider the retained of the two bins to be recovered from both AAMB and VAMB. From the de-replicated AVAMB bins, we ran GTDB-tk[36] v.2.1.0 on the bins both to assign taxonomies to each bin, and to obtain a multiple sequence alignment, from which we inferred a phylogenetic tree under the LG amino acid substitution model using IQ-TREE[37] v1.6.8. We annotated and visualised the tree with iTOL[38].

**Single-copy genes-based re-clustering**. Single-copy genes information can be used to postprocess AAMB and VAMB bins. After running AAMB(z) and VAMB, latents of bins with a mean number of single-copy genes greater than one are re-clustered using the weighted k-means clustering algorithm from SemiBin2[21]. Re-clustered bins replace the corresponding original AAMB and VAMB bins and are further de-replicated with the de-replication workflow (see De-replicating genomes between latent spaces sub-section).

**Statistics and reproducibility**. We found the de-replication pipeline to be completely reproducibly regarding the number of NC bins, composition, and quality scores, whereas we observed some variation on the bins' ids defined after processing AAMB and VAMB latent spaces.

**Reporting summary**. Further information on research design is available in the Nature Portfolio Reporting Summary linked to this article.

## Acknowledgements
We would like to acknowledge Nicolas Rascovan for his insights on the de-replication workflow interface. P.P., J.J., J.N.N., A.I.S and S.R. were supported by the Novo Nordisk Foundation grant NNF14CC0001. Furthermore, P.P., J.N.N., and S.R. were supported by the Novo Nordisk Foundation grant NNF20OC0062223. In addition, S.R. and S.K. were supported by the Novo Nordisk Foundation grant NNF19SA0059348. Finally, this work was also supported by the Novo Nordisk Foundation grant NNF21SA0072102.

## Author contributions
S.R. conceived the study and guided the analysis. P.P. developed AAMB and AVAMB and wrote the software and performed the analysis. Additionally, J.N.N. also performed analyses and wrote the software. J.J. and A.I.S. provided guidance and input for the analysis. S.K. contributed to the analysis of re-clustering. P.P., J.N.N., and S.R. wrote the manuscript with contributions from all co-authors. All authors read and approved the final version of the manuscript.

## Competing interests
The authors declare no competing interests.

## Data availability
The sequence data used in this study are publicly available from the respective studies or ENA. The semisynthetic MetaHIT dataset was downloaded from https://portal.nersc.gov/dna/RD/Metagenome_RD/MetaBAT/Files/ as the files depth.txt.gz and assembly-filtered.fa.gz. The simulated CAMI2 datasets were downloaded from https://data.cami-challenge.org/participate from '2nd CAMI Toy Human Microbiome Project Dataset'. The Almeida de novo assemblies were downloaded from http://ftp.ebi.ac.uk/pub/databases/metagenomics/umgs_analyses/benchmarked_assemblies.tar.gz and the reads were downloaded from ENA as specified in their publication. The HMP2 data was originally obtained from the European Nucleotide Archive accession PRJNA398089 and assembled as described in Johansen et al., 2022. Source data for the graphs in the main figures are available as Supplementary Data 1.

## Code availability
AVAMB is freely available at https://github.com/RasmussenLab/vamb/tree/master/workflow_avamb. AVAMB can also be installed and tested from Zenodo[39].

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
