## [Peer Review File · Communications Biology]

This manuscript has been previously reviewed at another Nature Portfolio journal. This document only contains reviewer comments and rebuttal letters for versions considered at Communications Biology.

REVIEWERS' COMMENTS:

Reviewer #3 (Remarks to the Author):

This paper developed a new binning method, "AAMB", to improve the performance of retrieving metagenome-assembled genomes from microbial communities. Contig binning is a vital processing step for reference-free metagenomic analysis. The use of adversarial deep autoencoder to generate latent spaces and the ensemble strategy for integrating bin sets are innovative. Having explored the current version of your software, I find it user-friendly and easy to run. However, I do have several major concerns regarding the manuscript:

1. While it is good that the authors compare AAMB with more recent binning methods like SemiBin2, it would be beneficial to also include Metadecoder in the comparison, as the first reviewer suggested. Their benchmarking results (<https://microbiomejournal.biomedcentral.com/articles/10.1186/s40168-022-01237-8/figures/6>) indicate Metadecoder as a powerful binning tool.
2. The binning performance of AAMB and AVAMB is not comparable to that of SemiBin and SemiBin2. Though the authors argue that AAMB is faster than SemiBin, I think the time consumption of SemiBin and SemiBin2 is still acceptable for biologists handling raw metagenomic data.
3. AVAMB relies on CheckM2 to do the dereplication and also employs CheckM2 to do the evaluation. I would like to suggest presenting both CheckM and CheckM2 evaluation results for AVAMB to ensure a more unbiased benchmarking process.

Minor:

4. The problem of reproducibility variations should be mentioned in the manuscript and providing potential suggestions would enhance the manuscript's credibility.
5. The authors only explore the taxa of MAGs retrieved by AVAMB and VAMB. Is it possible to extend such comparisons to more binning tools? For example, the authors can explore the species diversity retrieved by different bidders to demonstrate the superiority of the new method.

Once more we would like to thank the reviewers for their time and efforts in reviewing our manuscript. We appreciate the comments and suggestions and have made efforts to further benchmark AVAMB against new binners both in terms of near-complete (NC) bins and taxonomic diversity. Namely, we now benchmark AVAMB against MetaDecoder in the CAMI datasets as well as the Almeida dataset, both in terms of NC results (**Figure 2c, Supplementary Table 13**) and taxonomic diversity (**Supplementary Table 16**). We also benchmark AVAMB taxonomic diversity against MetaBAT2 on the Almeida dataset (**Supplementary Table 15**). In addition, besides evaluating AVAMB performance with CheckM, we reason for our decision to use CheckM2 to evaluate the final AVAMB bins. Finally, we evaluated the contribution of single-copy genes (SCG) to the performance of SemiBin2 as well as its impact when included in AVAMB. Interestingly, we found that SCG information contributes strongly to the performance of SemiBin2 and that AVAMB had better performance compared to SemiBin2 when compared with and without SCGs (**Supplementary Tables 12-13**).

Point-by-point response

Rev #1:

Major:

(1) 1. While it is good that the authors compare AVAMB with more recent binning methods like SemiBin2, it would be beneficial to also include Metadecoder in the comparison, as the first reviewer suggested. Their benchmarking results

(<https://microbiomejournal.biomedcentral.com/articles/10.1186/s40168-022-01237-8/figures/6>)

indicate Metadecoder as a powerful binning tool.

Author response:

We understand that comparing against other binners such as MetaDecoder could be beneficial. On the other hand, considering that SemiBin2 and MetaDecoder leverage single-copy genes for binning, we believe that a direct comparison against AVAMB is potentially misleading since AVAMB is only based on tetra-nucleotide frequencies and co-abundances to generate the bins. Therefore, we benchmarked AVAMB against SemiBin2 and MetaDecoder over the CAMI2 benchmark datasets where the SemiBin2 single-copy genes-based reclustering was applied to the AVAMB workflow (**Figure 2c, Supplementary Table 13**). These results showed that AVAMB + re-clustering was better compared to MetaDecoder and SemiBin2.

Figure 2. AAMB and VAMB reconstructed genomes analysis and integration with dereplication workflow. A. Jaccard correlation between the NC genomes from all benchmark datasets produced by AAMB(z), AAMB(y), VAMB, and AAMB(z+y) dereplicated bins. B. Contribution and the intersection of AAMB and VAMB for NC genomes on CAMI2 and MetaHIT datasets. The aggregated height of each bar expresses the total number of NC genomes reconstructed by VAMB and AAMB when dereplicating. Unique: NC genomes only reconstructed by the given binner; and: NC genomes reconstructed by all the binner connected with the and operator and not by any other binner; NC: Near complete. C. Number of distinct NC genomes reconstructed from the five CAMI2 benchmark datasets for AVAMB + RC (purple), SemiBin2 (dark grey), and MetaDecoder (light grey). Gi, Gastrointestinal; Urog, Urogenital; AVAMB + RC, AVAMB NC genomes when the single-copy genes-based re-clustering from SemiBin2 was applied to AVAMB's workflow. D. Comparison of strains, species, and genera recovered at NC level by VAMB (blue), AAMB(z) (light green), AAMB(y) (dark green), as well as AAMB(z+y) (light purple), and AVAMB (dark purple). S.e. of Ctenarytaina euc., secondary endosymbiont of Ctenarytaina eucalypti.

Dataset	AVAMB+RC	SemiBin2	MetaDecoder
Airways	102	96	94
Gi	145	140	127
Oral	165	159	140
Skin	143	138	133
Urog	111	112	104
Total	666	645	598

Supplementary Table 13. Near complete genomes reconstructed from the CAMI2 datasets by AVAMB bins re-clustered with SemiBin2 single-copy genes-based re-clustering implementation, SemiBin2 and MetaDecoder.

GI; Gastrointestinal, Urog; Urogenital. AVAMB + RC, AVAMB NC genomes when the single-copy genes-based re-clustering from SemiBin2 was applied to AVAMB's workflow.

Lines: 205-217 in the main text:

Namely, we benchmarked AVAMB against SemiBin (20), SemiBin2 (21), and MetaDecoder (13), and found that they reconstructed 15-36%, 15-45% and 1%-28% more NC genomes compared to AVAMB on the CAMI data sets respectively (**Supplementary Table 9**). In the case of SemiBin and Semibin2, performance increase came at a higher computational cost, taking 35-57 and 14-29 times longer than AVAMB respectively (**Supplementary Table 10**), whereas MetaDecoder running time was similar than AVAMB (**Supplementary Table 11**). Furthermore, we found SemiBin2 quite dependent on the single-copy genes-based re-clustering, given that AVAMB produced 0-32% more NC genomes on the CAMI data sets compared to SemiBin2 when single-copy genes-based re-clustering was not applied (**Supplementary Table 12**). In addition, when the single-copy genes-based re-clustering from SemiBin2 was applied to the AVAMB workflow, AVAMB outperformed SemiBin2 and MetaDecoder on 4/5 and 5/5 CAMI2 datasets respectively, reconstructing 21 and 68 more NC genomes along all CAMI2 datasets, respectively (**Figure 2c, Supplementary Table 13**).

(2) The binning performance of AAMB and AVAMB is not comparable to that of SemiBin and SemiBin2. Though the authors argue that AAMB is faster than SemiBin, I think the time consumption of SemiBin and SemiBin2 is still acceptable for biologists handling raw metagenomic data.

Author response:

We agree that the performance of SemiBin2 was superior to that of AVAMB. Nevertheless, we found the performance of SemiBin2 is highly conditioned by single-copy genes-based re-clustering. To illustrate the re-clustering impact, we applied the SemiBin2 re-clustering algorithm to AVAMB bins, showing that when re-clustering is applied both for SemiBin2 and AVAMB, AVAMB performance is superior in 4/5 datasets (**Figure 2c, Supplementary Table 13**). In addition, we included SemiBin2 performance over the CAMI datasets when no re-clustering was performed (**Supplementary Table 12**) where SemiBin2 without re-clustering had much lower performance compared to AVAMB (0-55% worse).

Dataset	AVAMB	SemiBin2 - no RC
Airways	82	62
Gi	103	103
Oral	138	130
Skin	104	67
Urog	83	73
Total	510	435

Supplementary Table 12. Near complete genomes reconstructed from the CAMI2 datasets by AVAMB and SemiBin2 when SemiBin2 single-copy genes-based re-clustering is not applied on SemiBin2 workflow. GI;

Gastrointestinal, Urog; Urogenital; SemiBin2 – no RC: SemiBin2 NC genomes without single-copy genes-based re-clustering.

Dataset	AVAMB+RC	SemiBin2	MetaDecoder
Airways	102	96	94
Gi	145	140	127
Oral	165	159	140
Skin	143	138	133
Urog	111	112	104
Total	666	645	598

Supplementary Table 13. Near complete genomes reconstructed from the CAMI2 datasets by AVAMB bins re-clustered with SemiBin2 single-copy genes-based re-clustering implementation, SemiBin2 and MetaDecoder. GI; Gastrointestinal, Urog; Urogenital. AVAMB + RC, AVAMB NC genomes when the single-copy genes-based re-clustering from SemiBin2 was applied to AVAMB's workflow.

Lines: 205-217 in the main text:

Namely, we benchmarked AVAMB against SemiBin (20), SemiBin2 (21), and MetaDecoder (13), and found that they reconstructed 15-36%, 15-45% and 1%-28% more NC genomes compared to AVAMB on the CAMI data sets respectively (**Supplementary Table 9**). In the case of SemiBin and Semibin2, performance increase came at a higher computational cost, taking 35-57 and 14-29 times longer than AVAMB respectively (**Supplementary Table 10**), whereas MetaDecoder running time was similar than AVAMB (**Supplementary Table 11**). Furthermore, we found SemiBin2 quite dependent on the single-copy genes-based re-clustering, given that AVAMB produced 0-32% more NC genomes on the CAMI data sets compared to SemiBin2 when single-copy genes-based re-clustering was not applied (**Supplementary Table 12**). In addition, when the single-copy genes-based re-clustering from SemiBin2 was applied to the AVAMB workflow, AVAMB outperformed SemiBin2 and MetaDecoder on 4/5 and 5/5 CAMI2 datasets respectively, reconstructing 21 and 68 more NC genomes along all CAMI2 datasets, respectively (**Figure 2c, Supplementary Table 13**).

(3) AVAMB relies on CheckM2 to do the dereplication and also employs CheckM2 to do the evaluation. I would like to suggest presenting both CheckM and CheckM2 evaluation results for AVAMB to ensure a more unbiased benchmarking process.

Author response:

We agree that using CheckM2 both during the dereplication and evaluation might introduce some biases during the process. Still, such biases mainly apply to the bins' selection process from AAMB(z), AAMB(y), and VAMB, whereas it has a marginal influence on the bins generation process. In addition, according to the lower performance of CheckM compared to CheckM2 on bins generated from CAMI2 (**Supplementary Figure 11**), we consider using CheckM for evaluation not optimal. In any case, we now report CheckM performance over AAMB, VAMB, and AVAMB bins generated from the Almeida and HMP2 datasets (**Supplementary Table 21**). An alternative could be to dereplicate

the bins with lower-quality CheckM2 bin scores. Nevertheless, due to the bottleneck that the dereplication process applies to the bins, reducing CheckM2 thresholds would increase the computational resources as well as compromise the final quality of the bins. Finally, we note that SemiBin and MetaDecoder leverage single-copy genes in the bin generation process whereas AVAMB only relies upon TNF and abundances to generate the bins and uses CheckM2 for the dereplication.

Dataset	AAMB(z)	AAMB(y)	VAMB	AAMB(z+y)	AAMB + VAMB
Almeida	4777	3221	5070	4751	5165
HMP2	3193	703	3122	2364	3095

Supplementary Table 21. CheckM NC bins reconstructed from the Almeida and Human Microbiome Project 2 datasets by AAMB(z), AAMB(y), VAMB, AAMB, and AVAMB. Near complete genomes according to CheckM reconstructed by AAMB(z), AAMB(y), VAMB, AAMB(z) and AAMB(y), VAMB and AAMB(z) and AAMB(y). Integration of binning results was done with the dereplication pipeline using CheckM2. AAMB(z): NC bins reconstructed from AAMB z latent space, AAMB(y): NC bins reconstructed from AAMB y latent space, AAMB(z+y): NC bins reconstructed from AAMB z latent space and AAMB y latent space, AAMB + VAMB: NC bins reconstructed from AAMB z latent space and AAMB y latent space and VAMB.

Lines 484-490 in the main text:

We acknowledge that using CheckM2 both during the dereplication process and the binning evaluation might introduce some biases. However, such biases mainly apply to the bins' selection process from AAMB(z), AAMB(y), and VAMB, rather than to the bin generation process. In addition, according to the lower performance of CheckM compared to CheckM2 on bins generated from CAMI2 (**Supplementary Figure 13**), we consider using CheckM for binning evaluation not optimal (**Supplementary Table 21**).

Minor:

(1) The problem of reproducibility variations should be mentioned in the manuscript and providing potential suggestions would enhance the manuscript's credibility.

Author response:

We appreciate the comment. However, considering that variation only occurs on the bin's names, not on the bin's composition nor in the quality scores, we believe that the reproducibility is acceptably robust.

Lines 490-493 on the main text:

Finally, we found the de-replication pipeline to be completely reproducibly regarding the number of NC bins, composition, and quality scores, whereas we observed some variation on the bins' ids defined after processing AAMB and VAMB latent spaces.

(2) The authors only explore the taxa of MAGs retrieved by AVAMB and VAMB. Is it possible to extend such comparisons to more binning tools? For example, the authors can explore the species diversity retrieved by different bidders to demonstrate the superiority of the new method.

Author response:

We recognize that AVAMB diversity was not extensively benchmarked. To address this matter, we now included the taxonomic diversity from running MetaBAT2 on the Almeida dataset (**Supplementary Table 19**), comparing it to AAMB and AVAMB reconstructed taxa. In addition, we also benchmarked the taxonomic diversity of AVAMB, AVAMB bins reclustered using SemiBin2 single-copy genes-based algorithm, and MetaDecoder on 30 randomly selected Almeida samples. (**Supplementary Table 20**).

Level	VAMB	AAMB	AVAMB	MetaBAT2
Domain	2	2	2	1
Phylum	13	13	13	13
Class	16	16	16	17
Order	40	39	41	38
Family	86	84	89	84
Genus	308	304	325	285
Species	677	707	767	605

Supplementary Table 19. We annotated all Almeida NC bins dereplicated bins from VAMB, AAMB, AVAMB and MetaBAT2 with GTDB-Tk and counted a particular taxon if at least one genome was reconstructed.

Level	AVAMB	AVAMB - rec	MetaDecoder
Domain	2	2	1
Phylum	9	9	7
Class	10	10	8
Order	17	17	14
Family	32	32	27
Genus	69	69	55
Species	102	102	69

Supplementary Table 20. We annotated all Almeida 30 samples NC bins AVAMB, AVAMB bins re-clustered with SemiBin2 single-copy genes-based re-clustering implementation and MetaDecoder with GTDB-Tk and counted

a particular taxon if at least one genome was reconstructed. AVAMB – rec; AVAMB bins re-clustered with SemiBin2 single-copy genes-based re-clustering implementation.

Lines 273-277 in the main text:

When benchmarking AVAMB against MetaBAT2, we found that AVAMB reconstructed 27% and 14% more unique species and genera than MetaBAT2 from the Almeida dataset (**Supplementary Table 19**). Finally, AVAMB performance was also higher than MetaDecoder on 30 randomly selected Almeida samples, yielding 48% and 25% more unique species and genera respectively (**Supplementary Table 20**).

Lines 249-255 in the main text:

Finally, we benchmarked AVAMB against MetaDecoder on 30 randomly selected samples from the Almeida dataset. On those samples, AVAMB reconstructed 148 NC genomes, whereas MetaDecoder reconstructed 82 NC genomes according to CheckM2. Besides the higher performance, AVAMB was 2.8 faster than MetaDecoder. AVAMB NC genomes performance did not increase when the single-copy genes-based re-clustering from SemiBin2 was applied to AVAMB bins.